# Risk Factors and Outcomes of Recurrent Candidemia in Children: Relapse or Re-Infection?

**DOI:** 10.3390/jcm8010099

**Published:** 2019-01-16

**Authors:** Mei-Yin Lai, Jen-Fu Hsu, Shih-Ming Chu, I-Hsyuan Wu, Hsuan-Rong Huang, Ming-Chou Chiang, Ren-Huei Fu, Ming-Horng Tsai

**Affiliations:** 1Division of Pediatric Neonatology, Department of Pediatrics, Chang Gung Memorial Hospital, Taoyuan 333, Taiwan; lmi818@msn.com (M.-Y.L.); jeff0724@gmail.com (J.-F.H.); kz6479@cgmh.org.tw (S.-M.C.); a9277@cgmh.org.tw (I.-H.W.); qbonbon@gmail.com (H.-R.H.); cmc123@cgmh.org.tw (M.-C.C.); rkenny@cgmh.org.tw (R.-H.F.); 2College of Medicine, Chang Gung University, Taoyuan 333, Taiwan; 3Division of Neonatology and Pediatric Hematology/Oncology, Department of Pediatrics, Chang Gung Memorial Hospital, Yunlin 333, Taiwan

**Keywords:** recurrent candidemia, late-onset sepsis, premature infants, antifungal resistance

## Abstract

In this paper, our aim was to investigate the incidence, clinical characteristics, risk factors, and outcomes of recurrent candidemia in children. We retrospectively reviewed all children with candidemia from a medical center in Taiwan between 2004 and 2015. Two episodes of candidemia ≥30 days apart with clinical and microbiological resolution in the interim were defined as “late recurrence”, and those that had 8–29 days apart from previous episodes were defined as “early recurrence”. 45 patients (17.2%) had 57 episodes of recurrent candidemia, and 24 had 28 episodes of late recurrent candidemia. The median time between recurrences was 1.8 months (range: <1 month to 13 months). Of those, 29 had relapsed candidemia and 28 were re-infected by different Candida species (*n* = 24) or by different strains (*n* = 4). Recurrent candidemia patients were more likely to require echinocandins treatment, had a longer duration of candidemia, and higher rate of treatment failure (*p* = 0.001, 0.014, and 0.012, respectively). Underlying gastrointestinal diseases (Odds ratio (OR) 3.84; 95% Confidence interval (CI) 1.81–8.12) and neurological sequelae (OR 2.32; 95% CI 1.15–4.69) were independently associated with the development of recurrent candidemia. 17.2% of pediatric patients with candidemia developed recurrent candidemia, and approximately half were re-infected. Underlying gastrointestinal diseases and neurological sequelae were the independent risk factors for recurrent candidemia.

## 1. Introduction

Candida species have become the fourth most common cause of nosocomial bloodstream infections since more than two decades ago [1]. Candidemia is associated with significant morbidity and contributes to important in-hospital mortality, especially among critically ill patients [2,3,4]. Most pediatric patients with candida bloodstream infections have severe underlying diseases, the presence of artificial devices, surgical risk factors, or are frequently exposed to high-risk medications [5,6,7,8,9]. There have been a number of studies that have focused on the epidemiology, antifungal susceptibility, clinical characteristics, risk factors, and outcomes of candidemia in children [5,6,7,8,9,10,11]. Currently, updated information of pediatric candidemia shows that emerging Candida isolates are nonsusceptible to fluconazole, a trend toward non-albicans Candida preponderance, and they still have high mortality [8,12,13,14,15].

Most studies on candidemia, whether from adult intensive care units (ICUs) or pediatric populations, almost always ignored the recurrent episodes of candidemia, and only retrieved the first episode of each patient for analysis [2,3,5,6,7,8,9,10,11]. Recurrent candidemia deserves greater concern, because most patients are still critically ill after surviving from the primary infection and may potentially have recurrent episodes [5,15,16,17]. These patients may have prolonged hospital stay, more severe illness, and persistent risk factors, which eventually lead to final mortality [15,16,17]. In the literature, recurrent candidemia in children has not been investigated. Furthermore, persistent candidemia and the occurrence of breakthrough candidemia have confused clinicians in regard to whether they were relapsed or re-infected [18,19,20]. In this study, we aimed to investigate recurrent episodes of candidemia in children and to document the incidence, clinical characteristics, and its impact on outcomes.

## 2. Patients and Methods

### 2.1. Study Design, Setting, and Ethics Approval

In a retrospective study, we examined the clinical microbiology database of Chang Gung Memorial Hospital (CGMH) to identify blood cultures that were positive for Candida spp. from all neonates and children ≤18 years of age from January 2004 to December 2015. CGMH is a tertiary level, university-affiliated teaching hospital in northern Taiwan. All Candida isolates were retrieved and re-identified using Matrix-assisted laser desorption ionization time-of-flight mass spectrometry (MALDI-TOF, Bruker Biotype, software version 3.0, Dallas, TX, USA) and large-subunit (18S) ribosomal RNA gene D1/D2 domain sequencing. We excluded unidentified Candida spp., and only enrolled candidemia with signs or symptoms of infection and ≥1 blood culture that was positive for Candida spp. This study and a waiver of informed consent for anonymous data collection were approved by the Institutional Review Board of the CGMH.

### 2.2. Definition and Data Collection

We defined an episode of candidemia defined as the first positive blood culture drawn from a peripheral vein that yields a Candida species, where the clinical symptoms and signs were compatible with Candida bloodstream infections (BSIs) [9,21]. Late recurrent candidemia was defined as the second episode of candidemia that occurred at least 30 days after the last positive blood culture positive for Candida [9,16,21,22]. For two different Candida species recovered within one week in the same patient, it was considered to be polyfungal candidemia. In cases of candidemia with more than two negative blood cultures, resolution of all septic symptoms and completion of antifungal treatment, the next positive blood culture yielding the same Candida species was sent to molecular diagnostics to document whether it was relapse or re-infection. Relapses were episodes caused by the same genus and species; re-infection was considered to be the second episode growing different Candida species after one week.

The clinical information was from the review of medical charts, and included demographic characteristics, predisposing risk factors within the preceding 30 days from the onset of Candida bloodstream infection (BSI) (defined as the day of first positive blood culture for Candida spp.), underlying diseases, and the presence of an intravenous catheter or any other artificial device at the time of candidemia.

### 2.3. Antifungal Susceptibility Testing

Antifungal susceptibility of all these Candida spp. isolates to eight antifungal agents was determined by a broth microdilution method, using the Sensititre YeastOne system (Trek Diagnostic Systems Ltd., East Grinstead, UK) according to the manufacturer’s instructions [23,24]. Minimum inhibitory concentration (MIC) was recorded as the highest concentration of antifungal agent resulting in the development of a blue color. The criteria for susceptibility of all Candida isolates to eight antifungal agents were based on MIC breakpoints of Candida spp. recommended by the Clinical and Laboratory Standards Institute (CLSI) [25]. For uncommon Candida spp., other than C. guilliermondii, clinical breakpoints are undefined; therefore, isolates that showed MICs higher than the epidemiologic cutoff value were considered potentially resistant [26].

### 2.4. Statistical Analysis

We compared the clinical characteristics, treatment, and outcomes of the first episode of candidemia with those of the recurrent episode of candidemia. The demographic, clinical, outcome variable, and in vitro susceptibility data were summarized using the descriptive statistics. All statistical analyses were performed using IBM SPSS software (version 22.0; IBM SPSS Inc., New York, NY, USA). Categorical variables were compared using the χ2 or Fisher’s exact test, and continuous variables by the Mann-Whitney U test. A *p*-value of 0.05 was considered significant.

## 3. Results

### 3.1. Recurrent and Late Recurrent Candidemia

Between January 2003 and December 2015, 262 patients less than 18 years old developed a total of 319 episodes of candidemia at CGMH (*C. albicans* 46.4%, *C. parapsilosis* 27.0%, *C. tropicalis* 5.6%, *C. glabrata* 5.0%, and other species 16.0%). In this cohort, 45 patients had a total of 57 episodes of recurrent candidemia. Of these, 28 episodes in 24 patients fulfilled the criteria of late recurrent candidemia. Seven patients had more than one recurrence, and four had a total of four episodes of candidemia. All recurrent episodes that had the same *Candida* species as the previous one were sent to molecular typing methods. We confirmed that 24 episodes were re-infected by different *Candida* spp., 29 episodes were relapsed by the same *Candida* strains and four episodes were re-infected by different strains of the same *Candida* spp. (Table 1). Of all pediatric patients with candidemia, 17.2% (45/262) had recurrent candidemia, and 9.2% (24/262) had late recurrent candidemia.

### 3.2. Characteristics of Recurrent Candidemia According to Time to Recurrence and Comparisons with First Episodes of Candidemia

Clinical characteristics of patients with recurrent candidemia and late recurrent candidemia are summarized in Table 2. Median time to early recurrence and late recurrence was 14 days (range, 10–29 days) and 72 days (range, 32–388 days), respectively. Cases of late recurrent candidemia and early recurrent candidemia were compared first, and then both were compared with the first episodes of candidemia. We found that the characteristics of late recurrent candidemia and early recurrent candidemia were mostly comparable (Appendix A). Patients with late recurrences were significantly younger (median, 1.8 years old) than patients with a single infection (median, 4.5 years; *p* = 0.013). When compared with the first episodes of candidemia, patients with recurrences had significantly more artificial devices (other than central venous catheter (CVC)), more previous azoles exposure, and previous bacteremia (all had *p* values <0.001). A large proportion of patients with recurrent candidemia had underlying gastrointestinal diseases (50.9%), neurological sequelae (50.9%), and other chronic comorbidities (Table 2). However, severity of illness and most underlying comorbidities were comparable in the first episodes and recurrent episodes of candidemia.

Most recurrent episodes were caused by non-*albicans Candida* species (70.2%); *C. parapsilosis,* especially was more likely to be the pathogen of recurrent candidemia than *C. albicans* (OR, 1.48; 95% CI 1.12–3.44, *p* = 0.020). Furthermore, 35.1% (19/57) of recurrent candidemia were breakthrough candidemia, and most of them were early re-infected by different *Candida* species during antifungal treatment.

### 3.3. Treatment and Outcomes of Recurrent Candidemia

Except for nine episodes of candidemia (all were first episodes), all other candidemia were treated with antifungal agents at a median of 2 days (range, 0–8 days) after onset of candidemia. Approximately two-thirds of first and recurrent episodes of candidemia were treated with fluconazole (65.3% vs. 68.4%, respectively), but the rate of antifungal regimens modification in recurrent episodes of candidemia was significantly higher than that in the first episodes. Besides, the recurrent episodes were more often treated with echinocandins than the first episodes (45.6% vs. 23.3%, *p* = 0.001) (Table 3). After modification of antifungal regimens, the recurrent episodes led to a significantly longer duration of candidemia (median 3.0 vs. 1.0 days, *p* = 0.014), slower antifungal agent responses, and significantly higher rates of clinical treatment failure (59.6% vs. 40.8%, *p* = 0.012). When we excluded cases that did not receive antifungal therapy (nine episodes in total), recurrent episodes of candidemia had a relatively higher candidemia-attributable mortality rate than the first episodes of candidemia (31.6% vs. 19.0%*, p* = 0.078).

### 3.4. Risk Factors of Recurrent Candidemia

Because of the similar characteristics of early recurrent candidemia and late recurrent candidemia, we enrolled all cases of recurrent candidemia and compared them with those with a single episode of candidemia. In non-neonatal children with candidemia, patients with recurrent candidemia were significantly younger than those with a single episode of candidemia (median age, 4.6 years vs. 2.5 years, *p* = 0.022). In univariate analysis, the characteristics of patients with recurrent candidemia were significantly more common, including underlying gastrointestinal diseases and neurological sequelae, presence of artificial devices other than CVC, and candidemia caused by *C. parapsilosis*. Disseminated candidiasis was highly associated with persistent candidemia, but was not associated with an increased risk of recurrent candidemia. Antifungal resistance of the first episode of candidemia was not significantly predisposed to the second episode, but delayed removal of CVC at the first episode was significantly associated with an increased risk of recurrent candidemia (odds ratio (OR) 2.33; 95% confidence interval (CI) 1.22–4.34; *p* = 0.022).

A logistic regression model was constructed to evaluate independent predictors for recurrent candidemia (Table 4). After adjusting for age and sex, underlying gastrointestinal diseases (OR 3.84; 95% CI 1.81–8.12; *p* = 0.001) and neurological sequelae (OR 2.32; 95% CI 1.15–4.69; *p* = 0.019) were significantly and independently associated with the development of recurrent candidemia. The Hosmer-Lemeshow goodness-of-fit test results indicated that the recurrent candidemia model reflected the data well (*p* = 0.88). 

## 4. Discussion

We found that 17.2% of pediatric patients who survived the first episode of candidemia had recurrent candidemia, and 9.2% had late recurrence. This result was significantly higher than previous studies that were conducted in an adult population, in which 1.5% to 4.4% of recurrent candidemia was reported [16,17,28]. We documented that more than half of all the recurrent episodes of candidemia were re-infections, where there was a higher probability of recurrence among patients with candidemia due to *C. parapsilosis*. Most of the characteristics and severity of illness were comparable between the first episodes and recurrent episodes. Although not statistically significant, the candidemia-attributable mortality rate in children with recurrent candidemia was higher than that in those with a single episode of candidemia.

Before investigating the issue of recurrent candidemia, it is important to differentiate cases of persistent candidemia. Most studies only focused on late recurrence, which was defined as an episode of candidemia that occurred at least one month after the last positive blood culture of previous fungemia [16,17,28,29,30]. In a recent study, only two episodes of candidemia ≥30 days apart with clinical, and microbiological resolution in the interim were enrolled [16]. In another study, a second episode caused by different *Candida* species (no mention of time interval) was considered as late recurrence [17]. However, we found that 22.8% (13/57) of recurrence were re-infection caused by different *Candida* species at 8–30 days after the previous episode, which was ignored in previous studies [16,17,28,29,30]. Although previous studies excluded cases of persistent candidemia, they may understate some episodes of early recurrent candidemia. Furthermore, we found that early re-infection candidemia had the highest candidemia-attributable mortality of 46.2%. Although our data was limited by the number of cases, as we are the first to consider early recurrent candidemia, the clinical significance remains a matter of debate.

In this study, we found that most characteristics of late recurrent candidemia and early recurrent candidemia were similar (Appendix A). We also found that re-infection and relapsed episodes were also comparable with regard to their characteristics. Therefore, our results can be compared with other studies of recurrent candidemia, which only enrolled episodes of late candidemia.

This study included all neonates and children with admissions in different wards or units, and some of their characteristics may be completely different. For example, immunodeficiency and autoimmune diseases were noted only in children. The treatment policies of data antifungals in children and neonates were also different. Therefore, all of their predisposing factors were considered in the analyses to identify independent risk factors for recurrence. Besides, the timeframe of the study period was long, and the antifungal options and guideline were changed during the study period (e.g. echinocandins were more widely used since 2007 in our institute). However, we previously documented that the treatment outcomes did not change over the study period [31]. Furthermore, although more non-albican *Candida* species, as well as antifungal resistant *Candida* isolates were noted in the recurrent episodes, initial antifungal agents did not independently affect the final outcomes [31].

The enrolment of early recurrent candidemia may explain the significantly higher rate of candidemia recurrence in our cohort. Furthermore, most pediatric patients survived their first episode of candidemia, and they were still at risk of subsequent candidemia, including numerous chronic comorbidities, immunocompromised status, and the presence of an intravenous catheter or other artificial devices. Therefore, the incidence of recurrent candidemia in children was significantly higher than that in adults [16,17]. We also found that most mortality in children with candidemia resulted from recurrent infection, subsequent infectious complications, and organ damage, which is consistent with our previous study [32].

Recent studies have found more recurrent candidemia to be re-infections [16,17], contrary to earlier studies that suggested that most recurrent episodes represent persistent, rather than recurrent infections [28,29,30]. In this study, re-infection accounted for 49.1% (28/57) of recurrence. The trend of more re-infections and less persistent (or relapsed) candidemia may be due to different treatment strategies, increasing non-*albicans* candidemia, and increasing *Candida* species that have antifungal resistance [22,33]. This is compatible with our result that non-*albicans* candidemia accounted for the majority of recurrent candidemia. We found most of our patients still had chronic comorbidities and risk factors after they survived the first episode of candidemia. These recurrences should be caused by the persistence of particular risk factors for candidemia rather than inadequately treated infections. Thus, this is the scenario where the pediatric patient with candidemia was first treated successfully, but the persistence of chronic underlying diseases makes him susceptible to another episode of candidemia caused by non-*albicans Candida* species after antifungal selection.

Among the risk factors for candidemia, underlying gastrointestinal sequelae were independently associated with recurrent candidemia after multivariate logistic regression, which is consistent with previous studies [16,17]. Both Muñoz et al. [16] and Asmundsdottir et al. [17] found that more than half of the patients with recurrent candidemia had an underlying gastrointestinal disease. In our cohort, gastrointestinal sequelae were the most common underlying chronic comorbidities in patients with recurrent candidemia (50.9%). Because Candida species commonly colonize in the human gastrointestinal tract as a component of the resident microbiota [27,34], it is possible that these microorganisms may serve as the sources of invasive infection during immunocompromised status and/or impaired gastrointestinal function.

Our result is consistent with previous studies that *C. parapsilosis* accounts for a significant proportion of recurrent candidemia [16,30]. In this study, we also found that all episodes of recurrent candidemia occurred in patients with CVC in place, and there was a significantly higher proportion of other artificial devices (71.9%) in these patients. Because *C. parapsilosis* is well-known for its ability to form biofilm on implanted devices [35,36], we suggested that the presence of artificial devices clotted with biofilm-candida complex may serve as another important source of recurrence. Further studies regarding the microbiological characteristics of biofilm-forming *Candida* isolates and new strategies to eradicate the residual microorganisms after initial candidemia may contribute to decreasing the incidence of recurrent candidemia.

In our institute, empirical antifungal agent is prescribed only when clinical candidiasis is suspected, and this depends on the physician’s decision. We did not have a specific policy of antifungal prophylaxis in the neonatal or pediatric intensive care unit. Because it is often difficult to distinguish candidemia from bacteremia in the beginning, clinicians are used to prescribe empiric antibiotics rather than empiric antifungal agents; therefore, a significant proportion of candidemia (approximately 4 out of 10) did not receive effective therapy initially. Furthermore, the removal of CVC is also decided by the clinicians in our institute. Although removal of CVC is critically important when candidemia is documented [31], it usually requires the availability of an alternative intravascular route and that the patient is in stable condition.

This study has some limitations. Because of its retrospective nature in a single center, and also because follow-up blood cultures after having documented candidemia were not drawn regularly, it was sometimes difficult to differentiate persistent candidemia from early relapsed candidemia. The conclusion of this current study is also limited by the retrospective design and heterogenous enrolment of the subjects studied. Although MALDI-TOF can rapidly identify the Candida spp. in a sensitive and economical way, a major disadvantage is that the spectral database must contain peptide mass fingerprints of the new specific species; otherwise, misdiagnosis may happen [37]. Furthermore, undocumented candidemia in patients on empiric antifungal therapy may lead to unrepresented cases in this study and the underestimated incidence of candidemia. Therefore, a prospective study which mandates detailed and continuous diagnostic work in order to define the recurrence of an episode is warranted.

## Figures and Tables

**Table 1 jcm-08-00099-t001:** Patients with recurrent candidemia: pathogens, characteristics, treatment, and outcomes.

Early and Late Recurrences	*Candida* Species of Initial Episodes	Days until Recurrence (Range)	*Candida* Species of Recurrent Episodes	MIC (μg/mL) to Fluconazole (Case Number)	Therapeutic Regimens	Candidemia- Attributable Mortality, *n* (%)
Early recurrence *						
Relapsed (*n* = 16)	*C. albicans* (7)	12–25	*C. albicans* (7)	0.5 (5), 16.0 (2)	Amphotericin b (4),	4 (25.0)
	*C. parapsilosis* (6)	12–22	*C. parapsilosis* (6)	0.25–2.0	Fluconazole (3),	
	*C. glabrata* (3)	12–20	*C. glabrata* (3)	16 (1), 32 (2)	Echinocandin (8), Flu + Echinocandins (1)	
Re-infection (*n* = 13)	*C. albicans* (8)	10–20	*C. parapsilosis* (6), *C. metapsilosis* (1), *C. tropicalis* (1)	0.5–1.0 (5), 8 (1) 2.0 4.0	Echinocandins (5), Fluconazole (2), Flu + Echinocandins (1)	6 (46.2)
	*C. parapsilosis* (1)	17	*C. glabrata* (1)	16.0	Amphotericin b (1)	
	*C. tropicalis* (2)	10–15	*C. lusitaniae* (1), *C. parapsilosis* (1)	0.25, 2.0	Echinocandins (1), Amphotericin b (1)	
	*C. metapsilosis* (1)	12	*C. lipolytica* (1)	2.0	Echinocandins (1)	
	*C. lusitaniae* (1)	11	*C. haemulonii* (1)	16.0	Echinocandins (1)	
Late recurrence *						
Relapsed (*n* = 13)	*C. albicans* (5)	49–101	*C. albicans* (5)	0.25 (1), 0.5 (3), 16 (1)	Echinocandins (2), Fluconazole (3)	4 (30.8)
	*C. parapsilosis* (5)	32–162	*C. parapsilosis* (5)	0.5–2.0	Amphotericin b (1), Echinocandins (3), Fluconazole (1)	
	*C. tropicalis* (1)	31	*C. tropicalis* (1)	2.0	Echinocandins (1)	
	*C. glabrata* (1)	34	*C. glabrata* (1)	16.0	Amphotericin b (1)	
	*C. orthopsilosis* (1)	59	*C. orthopsilosis* (1)	2.0	Echinocandins (1)	
Re-infection (*n* = 15)	*C. albicans* (7)	45–155	*C. albicans* (2), *C. parapsilosis* (3), *C. guilliermondii* (1), *C. tropicalis* (1)	0.5 0.5–1.0 4.0 2.0	Fluconazole (2), Amphobericin b (2), Echinocandins (3)	4 (26.7)
	*C. parapsilosis* (3)	54–136	*C. parapsilosis* (1), *C. glabrata* (1), *C. metapsilosis* (1).	0.25 8.0 2.0	Fluconazole (2), Amphotericin b (1)	
	*C. glabrata* (2)	31–72	*C. glabrata* (1), *C. parapsilosis* (1)	8.0, 0.5	Fluconazole (1), Amphotericin b (1)	
	*C. tropicalis* (2)	65–114	*C. albicans* (2)	0.5, 2.0	Amphotericin b (2)	
	*C. metapsilosis* (1)	388	*C. albicans* (1)	0.5	Fluconazole (1)	

* Early recurrence indicated the recurrence occurred within 8–29 days after the last positive blood culture of first episode of candidemia, whereas late recurrence indicated recurrence occurred at least 30 days later.

**Table 2 jcm-08-00099-t002:** Comparisons of first episodes and recurrent episodes of candidemia in children.

Characteristics	First Episodes of Candidemia (*n* = 262)	Recurrent Episodes of Candidemia (Total *n* = 57)			
		All Recurrence (*n* = 57)	*p* Value *	Late Recurrence (*n* = 28)	*p* Value **
Age					
Non-neonatal patients (years), median (IQR)	4.5 (1.4–2.1)	2.9 (0.9–7.2)	0.059	1.8 (0.8–7.1)	0.013
Neonatal patients (days), median (IQR)	22.5 (14.8–44.3)	57.5 (39.5–79.5)	<0.001	66.5 (52.5–76.3)	<0.001
Sex (male gender)	138 (52.7)	27 (47.4)	0.559	13 (46.4)	0.556
Hospital days untilΙ diagnosis, median (IQR)	26.0 (13.0–45.0)	43.0 (28.0–76.5)	<0.001	44.0 (19.5–84.5)	<0.001
Ward			0.056		0.059
Neonatal intensive care unit	94 (35.9)	12 (21.1)		5 (17.9)	
Pediatric intensive care unit	95 (36.3)	29 (50.9)		12 (42.9)	
Burn or surgical intensive care unit	8 (3.1)	5 (8.8)		3 (10.7)	
General wards	65 (24.8)	11 (19.3)		8 (28.6)	
Underlying chronic comorbidities ^#^					
Congenital or genetic anomalies	28 (9.9)	11 (19.3)	0.078	5 (17.9)	0.256
Neurological sequelae	89 (34.0)	29 (50.9)	0.023	17 (60.7)	0.005
Cardiovascular disease	22 (8.4)	9 (15.8)	0.077	3 (10.7)	0.720
Chronic lung disease and/or pulmonary hypertension	82 (31.3)	19 (33.3)	0.756	6 (21.4)	0.387
Gastrointestinal sequelae	71 (27.1)	29 (50.9)	0.001	17 (60.7)	0.001
Renal insufficiency with/without dialysis	37 (14.1)	8 (14.0)	1.000	4 (14.3)	1.000
Hematological/Oncology cancer	42 (16.0)	4 (7.0)	0.079	0 (0)	0.020
Immunodeficiency	6 (2.3)	1 (1.8)	0.802	0 (0)	0.541
Autoimmune disease	7 (2.7)	1 (1.8)	0.688	0 (0)	0.487
Hepatic failure or cholestasis	12 (4.6)	1 (1.8)	0.328	1 (3.6)	1.000
Pathogens			0.012		0.806
*Candida albicans*	131 (50.0)	17 (29.8)	0.008	10 (35.7)	
*Candida parapsilosis*	63 (24.0)	23 (40.4)	0.020	10 (35.7)	
*Candida tropicalis*	17 (6.5)	3 (5.3)		2 (7.1)	
*Candida glabrata*	12 (4.6)	7 (12.3)		3 (10.7)	
*Candida guilliermondii*	11 (4.2)	1 (1.8)		1 (3.6)	
Other *Candida* spp.	28 (10.7)	6 (10.5)		2 (7.1)	
Clinical presentation					
Severe sepsis	102 (38.9)	25 (43.9)	0.551	10 (35.7)	0.840
Septic shock	69 (26.3)	20 (35.1)	0.194	8 (28.6)	0.823
Progressive and deteriorated ^¶^	47 (17.9)	14 (24.6)	0.266	4 (14.3)	0.797
Disseminated candidiasis ^$^	12 (4.6)	2 (3.5)	0.720	1 (3.6)	1.000
Breakthrough candidemia	20 (7.6)	20 (35.1)	<0.001	5 (17.9)	0.067
Predisposing risk factors ^#^					
Receipt of systemic antibiotics ^&^	262 (100)	56 (98.2)	0.142	27 (96.4)	0.705
Previous azole exposure ^&^	10 (3.8)	24 (42.1)	<0.001	9 (32.1)	<0.001
Prior bacteremia ^&^	114 (43.5)	46 (80.7)	<0.001	21 (75.0)	0.002
Presence of CVC	253 (96.6)	57 (100)	0.371	28 (100)	0.319
Stay in an intensive care unit	197 (75.2)	46 (80.7)	0.565	16 (57.1)	0.080
Receipt of parenteral nutrition	175 (66.8)	42 (73.7)	0.350	21 (75.0)	0.378
Receipt of immunosuppressants	55 (21.0)	10 (17.5)	0.717	2 (7.1)	0.085
Artificial device other than CVC	119 (45.4)	41 (71.9)	<0.001	21 (75.0)	0.003
Prior surgery ^&^	80 (30.5)	19 (33.3)	0.753	7 (25.0)	0.528
Neutropenia (ANC < 0.5 × 10^3^/μL)	63 (24.0)	17 (29.8)	0.400	6 (21.4)	1.000

* *p* Values were the comparisons between all first episodes of candidemia and all recurrent episodes of candidemia. ** *p* Values were the comparisons between all first episodes of candidemia and all late-recurrent episodes of candidemia. ^#^ Indicated the presence of an underlying condition or risk factor at the onset of candidemia, and most patients with candidemia had >1 underlying condition and/or risk factor. ^&^ Within one month prior to the onset of candidemia, prior azoles exposure indicated patients received the azoles drug in addition to antifungal agents at the time of candidemia. ^¶^ Defined as candidemia episodes with more disseminated candidiasis and/or progressive multi-organ failure even after effective antifungal agents. ^$^ Indicated positive *Candida* isolates recovered from more than two sterile sites, in addition to primary bloodstream infection. IQR: interquartile range; CVC: central venous catheter; ANC: absolute neutrophil count.

**Table 3 jcm-08-00099-t003:** Antifungal regimens and outcomes of recurrent episodes of candidemia versus first episodes of candidemia.

	Recurrent Episodes of Candidemia (total *n* = 57)	First Episodes of Candidemia (Total *n* = 262)	*p* Value
Duration of candidemia (days), median (interquartile range)	3.0 (1.0–9.0)	3.0 (1.0–6.0)	0.638
≤2 days	26 (45.6)	115 (43.9)	
3–7 days	14 (24.5)	101 (38.5)	
≥8 days	17 (29.8)	46 (17.6)	
Antifungal regimens for treatment			0.007
Fluconazole/Voriconazole	15 (26.3)	107 (40.8)	0.050
Amphotericin B	14 (24.6)	80 (30.5)	
Echinocandins	26 (45.6)	61 (23.3)	0.001
Combination antifungal treatment	2 (3.5)	5 (1.9)	
None	0 (0)	9 (3.4)	
Effective antifungal agents given within 48 hours after onset of candidemia (based on antifungal susceptibility testing)	28/57 (49.1)	162/262 (61.8)	0.102
Total treatment duration (days), mean (interquartile range)	21.0 (14.0–25.0)	16.0 (14.0–22.0)	0.338
Catheter removal	33/57 (57.9)	176/262 (67.2)	0.108
Removal of central venous catheter within 3 days of onset	16/57 (28.1)	105/262 (40.1)	0.082
Treatment outcomes			
Responsiveness after initiation of antifungal treatment *			0.014
Within 72 h	13 (22.8)	109 (41.6)	
4–7 days	9 (15.8)	51 (19.5)	
More than 7 days	16 (28.1)	42 (16.0)	
Treatment failure	34 (59.6)	107 (40.8)	0.012
Modification of antifungal treatment	32/57 (56.1)	104/253 (41.1)	0.054
Duration of candidemia after effective antifungal agents (days), median (IQR)	3.0 (1.0–7.5)	1.0 (1.0–4.0)	0.014
Candidemia attributable mortality	18 (31.6)	57 (21.8)	0.123
Early mortality (≤7 days)	8 (14.0)	27 (10.3)	0.482
Late mortality (8–30 days)	10 (17.5)	30 (11.5)	0.267

All data have been expressed as a number (percentage %), unless indicated otherwise. * Responsiveness to antifungal agents was defined according to the consensus criteria of the Mycoses Study Group and European Organization for Research and Treatment of Cancer [27].

**Table 4 jcm-08-00099-t004:** Multivariate analysis to identify the independent risk factors of recurrent candidemia.

Variables	Odds Ratio	95% Confidence Interval	*P*
Male sex	1.05	0.54–2.05	0.889
Age (neonates vs. children)	1.58	0.68–3.65	0.286
*Candida parapsilosis*	2.57	0.89–7.42	0.080
Gastrointestinal underlying diseases	3.84	1.81–8.12	0.001
Underlying neurological sequelae	2.32	1.15–4.69	0.019
Underlying cardiovascular diseases	2.24	0.85–6.69	0.090
Presence of artificial devices other than central venous catheter	1.73	0.84–3.54	0.136
Catheter management of first episode of candidemia (delayed removal vs. early removal)	1.12	0.68–1.13	0.302

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
