# Peer review of "Risk Factors and Outcomes of Recurrent Candidemia in Children: Relapse or Re-Infection?"

_jcm, 2019, doi:10.3390/jcm8010099_

Round 1
Reviewer 1 Report
In the manuscript entitled "Risk factors and outcomes of recurrent candidemia in children: relapse or re-infection" the authors present a retrospective data analysis of 262 children concering the rates of recurrent candidemia. Although the data are presented well the scientific quality is limited due to the fact that a small retrospective study is submitted.
Introduction: There is some publication on recurrent candidaemia in children [e.g. Factors and outcomes associated with candidemia caused by non-albicans Candida spp versus Candida albicans in children. Am J Infect Control. 2018 Dec;46(12):1387-1393. ] Please, at least, update the introduction part of the manuscript concerning the review of the literature. In the discussion part of the manuscript several trials on candidiasis (in adult patients) are discussed in an adequate manner. Please focus on the disadvantages of these trials concering the “new” detection method you used.
Despite the importance of the topic for pediatric patients the study group enrolled into the trial is very heterogenous. In cases of a major revision the discussion should focus on the disadvantages of this retrospective analysis.
The conclusion is adequate and carefully summarizes the results of this trial.
I would recommend the manuscript for revision.
Author Response
RE: jcm-417111 Journal of Clinical Medicine
Title: Risk factors and outcomes of recurrent candidemia in children: relapse or re-infection ?
Dear reviewer,
Thank you for your appreciated comments on our manuscript. I will have the manuscript revised, all according to the reviewers’ and editor’s suggestions. We underline every change and highlight in red color on the revised manuscript. The replies for the reviewers’ criticisms are as followings. We hope this revised version can be acceptable. I appreciate your further reviews and comments, thank you.
Best regards,
Ming-Horng Tsai
Chief, Division of Neonatology and Pediatric Hematology/Oncology, Department of Pediatrics, Yunlin Chang Gung Memorial Hospital, Taiwan, R.O.C.
In the manuscript entitled "Risk factors and outcomes of recurrent candidemia in children: relapse or re-infection" the authors present a retrospective data analysis of 262 children concering the rates of recurrent candidemia. Although the data are presented well the scientific quality is limited due to the fact that a small retrospective study is submitted.
Reply:
Thank you for your review and comments. I have described the scientific quality is limited due to the fact that this is a small retrospective study in the last paragraph of the discussion. Therefore, further multicenter study is warranted in the future, thank you.
Introduction: There is some publication on recurrent candidaemia in children [e.g. Factors and outcomes associated with candidemia caused by non-albicans Candida spp versus Candida albicans in children. Am J Infect Control. 2018 Dec;46(12):1387-1393. ] Please, at least, update the introduction part of the manuscript concerning the review of the literature. In the discussion part of the manuscript several trials on candidiasis (in adult patients) are discussed in an adequate manner. Please focus on the disadvantages of these trials concerning the “new” detection method you used.
Reply:
Thank you for your instructive advice. I will update the introduction part of the manuscript concerning the review of the literature. I will add the above reference in the introduction section (reference no.15). For the discussion section, I will describe more disadvantages of these trials concerning the “new” detection method I used (in Page 17, the last paragraph of discussion section: Although MALDI-TOF can rapidly identify the Candida spp. in a sensitive and economical way, the major disadvantage is that the spectral database must contain peptide mass fingerprints of the new specific species. Otherwise, misdiagnosis may happen [37].)
Despite the importance of the topic for pediatric patients the study group enrolled into the trial is very heterogenous. In cases of a major revision the discussion should focus on the disadvantages of this retrospective analysis.
Reply:
Thank you for your instructive advice. I will focus on the disadvantages of this retrospective analysis. I will add this issue in the last paragraph of discussion section, (page 17 the last paragraph of discussion).
The conclusion is adequate and carefully summarizes the results of this trial.
I would recommend the manuscript for revision.
Reply:
Thank you for your review and comments.
Reviewer 2 Report
The study “Risk factors and outcomes of recurrent candidemia in children: relapse or re-infection?” is a retrospective 12-year observational study of recurrent episodes of candidemia in neonates and children treated in a tertiary, university-affiliated teaching hospital of Northern Taiwan.
The study is very interesting because, as the authors acknowledge, the available literature on pediatric recurrent candidemia is limited. The authors need to be congratulated because they retrospectively retrieved and re-identified all clinical Candida isolates implicated in cases of candidemia using MALDI-TOF and 18S ribosomal RNA gene D1/D2 domain sequencing. They showed that among 57 episodes of candidemia, approximately half (50.9%) were relapses and the remaining were re-infections by other Candida spp, or by different strains of the same Candida spp.
Criticism
1. A main finding of the study is that underlying gastrointestinal diseases and neurological sequelae were independently associated with development of recurrent candidemia. This should be included in the conclusion of the abstract. This is a main finding of the study and should be certainly included in the abstract along with the finding that 17.2% of children with candidemia develop recurrent candidemia, half of which was due to re-infection.
2. In Table 3, the authors show that effective antifungal agents were given within 48 hours after the onset of candidemia in only 61.8% of the first episodes of candidemia. This is a low percentage, implying that approximately 4 out of 10 children received ineffective therapy initially. The authors should comment on that and briefly discuss what their policy was regarding antifungal prophylaxis and empirical antifungal therapy during the 12-year study period. I realize that neonates and children from different wards and units were included over a long time (more than a decade), but still a brief discussion of antifungal policies should be included.
3. In Table 3, it is also shown that the CVCs were more easily removed in first episodes of candidemia than in recurrent ones, which does not make a lot of sense. Although this was marginally not significant (p=0.082), the authors should comment on their policy regarding catheter removal in the discussion.
4. The main finding of the study, i.e., the association of underlying gastrointestinal diseases with development of recurrent candidemia makes a lot of sense, since the usual source of candidemia in critically ill and immunocompromised patients is the gut, although this is a matter of debate (Clin Infect Dis. 2001;33: 1959‐67). The authors should further comment on that in the discussion of the revised manuscript.
5. The definition of early recurrence in the abstract (candidemia occurring 8-29 days from previous episode) is different from the definition given in Table 1 (*Early recurrence indicated the recurrence occurred within 8-30 days after the last positive blood culture of first episode of candidemia). Please, be consistent. Obviously late recurrence is ≥30 days, but this should be consistently applied.
6. By carefully looking the data of Table 1, it seems that only in 2 cases was the re-infection due to the same species (but different strains), one case of late re-infection by C. albicans and another case of late re-infection by C. glabrata. I do not see the remaining 2 cases of re-infection by the same Candida spp., since the authors claim in page 6, line 147 that 4 episodes were re-infection by different strains of the same Candida spp. Please, explain or make a special note for these 4 episodes in detail.
7. Because the number of patients (n=45) is different from the number of episodes of recurrent candidemia (n=57), as some patients had >1 episode, Table 1 is confusing and should be re-constructed including the number of patients. Alternatively, another Table should be made including the patients with >1 episode of candidemia.
8. The manuscript’s needs some limited editing from a native English speaker in order to improve its readability.
Author Response
RE: jcm-417111 Journal of Clinical Medicine
Title: Risk factors and outcomes of recurrent candidemia in children: relapse or re-infection ?
Dear reviewer,
Thank you for your appreciated comments on our manuscript. I will have the manuscript revised, all according to the reviewers’ and editor’s suggestions. We underline every change and highlight in red color on the revised manuscript. The replies for the reviewers’ criticisms are as followings. We hope this revised version can be acceptable. I appreciate your further reviews and comments, thank you.
Best regards,
Ming-Horng Tsai
Chief, Division of Neonatology and Pediatric Hematology/Oncology, Department of Pediatrics, Yunlin Chang Gung Memorial Hospital, Taiwan, R.O.C.
The study “Risk factors and outcomes of recurrent candidemia in children: relapse or re-infection?” is a retrospective 12-year observational study of recurrent episodes of candidemia in neonates and children treated in a tertiary, university-affiliated teaching hospital of Northern Taiwan.
The study is very interesting because, as the authors acknowledge, the available literature on pediatric recurrent candidemia is limited. The authors need to be congratulated because they retrospectively retrieved and re-identified all clinical Candida isolates implicated in cases of candidemia using MALDI-TOF and 18S ribosomal RNA gene D1/D2 domain sequencing. They showed that among 57 episodes of candidemia, approximately half (50.9%) were relapses and the remaining were re-infections by other Candida spp, or by different strains of the same Candida spp.
Criticism
1. A main finding of the study is that underlying gastrointestinal diseases and neurological sequelae were independently associated with development of recurrent candidemia. This should be included in the conclusion of the abstract. This is a main finding of the study and should be certainly included in the abstract along with the finding that 17.2% of children with candidemia develop recurrent candidemia, half of which was due to re-infection.
Reply:
Thank you for your instructive advice. I will add this issue in the conclusion of the abstract (the last sentence), thank you.
2. In Table 3, the authors show that effective antifungal agents were given within 48 hours after the onset of candidemia in only 61.8% of the first episodes of candidemia. This is a low percentage, implying that approximately 4 out of 10 children received ineffective therapy initially. The authors should comment on that and briefly discuss what their policy was regarding antifungal prophylaxis and empirical antifungal therapy during the 12-year study period. I realize that neonates and children from different wards and units were included over a long time (more than a decade), but still a brief discussion of antifungal policies should be included.
Reply:
Thank you for your instructive advice. In our institute, we did not have antifungal prophylaxis. I will discuss this issue in the discussion section, page 16, the last paragraph: In our institute, empirical antifungal agent is prescribed only when clinical candidiasis is suspected and this depends on the physician’s decision. We did not have specific policy of antifungal prophylaxis in the neonatal or pediatric intensive care unit. Because it is often difficult to distinguish candidemia from bacteremia in the beginning, clinicians are used to prescribe empiric antibiotics rather than empiric antifungal agents; therefore, a significant proportion of candidemia (approximately 4 out of 10) did not receive effective therapy initially. Furthermore, the removal of CVC is also decided by the clinicians in our institute. Although removal of CVC is critically important when candidemia is documented [30], it usually requires availability of alternative intravascular route and the patient is in stable condition.
3. In Table 3, it is also shown that the CVCs were more easily removed in first episodes of candidemia than in recurrent ones, which does not make a lot of sense. Although this was marginally not significant (p=0.082), the authors should comment on their policy regarding catheter removal in the discussion.
Reply:
Thank you for your instructive advice. I will comment on the policy regarding catheter removal when candidemia was diagnosed in the discussion section. (page 17, first paragraph) Furthermore, the removal of CVC is also decided by the clinicians in our institute. Although removal of CVC is critically important when candidemia is documented [30], it usually requires availability of alternative intravascular route and the patient is in stable condition.
4. The main finding of the study, i.e., the association of underlying gastrointestinal diseases with development of recurrent candidemia makes a lot of sense, since the usual source of candidemia in critically ill and immunocompromised patients is the gut, although this is a matter of debate (Clin Infect Dis. 2001;33: 1959‐67). The authors should further comment on that in the discussion of the revised manuscript.
Reply:
Thank you for your instructive advice. This issue has been discussed in the discussion section, page 15, the last paragraph until page 16, the first paragraph.
5. The definition of early recurrence in the abstract (candidemia occurring 8-29 days from previous episode) is different from the definition given in Table 1 (*Early recurrence indicated the recurrence occurred within 8-30 days after the last positive blood culture of first episode of candidemia). Please, be consistent. Obviously late recurrence is ≥30 days, but this should be consistently applied.
Reply:
Thank you for your instructive advice. I will revise the early recurrence as those occurred within 8-29 days after the last positive blood culture of the first episode (in table 1).
6. By carefully looking the data of Table 1, it seems that only in 2 cases was the re-infection due to the same species (but different strains), one case of late re-infection by C. albicans and another case of late re-infection by C. glabrata. I do not see the remaining 2 cases of re-infection by the same Candida spp., since the authors claim in page 6, line 147 that 4 episodes were re-infection by different strains of the same Candida spp. Please, explain or make a special note for these 4 episodes in detail.
Reply:
Thank you for your instructive advice. I am sorry for this mistake. I checked the initial data. When I transferred the initial data to the Table 1, I put one recurrent episode caused by C. albicans and one caused by C. parapsilosis in the wrong place. I have corrected them and revised in the Table 1, thank you.
7. Because the number of patients (n=45) is different from the number of episodes of recurrent candidemia (n=57), as some patients had >1 episode, Table 1 is confusing and should be re-constructed including the number of patients. Alternatively, another Table should be made including the patients with >1 episode of candidemia.
Reply:
Thank you for your instructive advice. Because Table 1 focused on numbers of episode instead of the patient, it is difficult to incorporate the number of patients into this table. For alternatively, if another Table is made including all the patients with > 1 episode of candidemia, this table will be very large (total more than 45 rows). I suggest not making this issue too confusing. If the reviewer still insisted me to do another table, please let me know, and I will try.
8. The manuscript’s needs some limited editing from a native English speaker in order to improve its readability.
Reply:
Thank you for your instructive advice. We have professional English editing from a native English speaker in our Institute. All the revised parts are in red color, thank you.
Round 2
Reviewer 1 Report
The manuscript under consideration was improved significantly in the revision process. The most recent literature is included now and the discussion is well-balanced.
I can recommend it for publication now.